# Profiling Persistent Asthma Phenotypes in Adolescents: A Longitudinal Diagnostic Evaluation from the INSPIRERS Studies

**DOI:** 10.3390/ijerph18031015

**Published:** 2021-01-24

**Authors:** Rita Amaral, Cristina Jácome, Rute Almeida, Ana Margarida Pereira, Magna Alves-Correia, Sandra Mendes, José Carlos Cidrais Rodrigues, Joana Carvalho, Luís Araújo, Alberto Costa, Armandina Silva, Maria Fernanda Teixeira, Manuel Ferreira-Magalhães, Rodrigo Rodrigues Alves, Ana Sofia Moreira, Ricardo M. Fernandes, Rosário Ferreira, Paula Leiria Pinto, Nuno Neuparth, Diana Bordalo, Ana Todo Bom, Maria José Cálix, Tânia Ferreira, Joana Gomes, Carmen Vidal, Ana Mendes, Maria João Vasconcelos, Pedro Morais Silva, José Ferraz, Ana Morête, Claúdia Sofia Pinto, Natacha Santos, Claúdia Chaves Loureiro, Ana Arrobas, Maria Luís Marques, Carlos Lozoya, Cristina Lopes, Francisca Cardia, Carla Chaves Loureiro, Raquel Câmara, Inês Vieira, Sofia da Silva, Eurico Silva, Natalina Rodrigues, João A. Fonseca

**Affiliations:** 1Center for Health Technology and Services Research (CINTESIS), Faculty of Medicine, University of Porto, 4200-319 Porto, Portugal; cristinajacome.ft@gmail.com (C.J.); rutealmeida@med.up.pt (R.A.); ambrpereira@gmail.com (A.M.P.); sandra.mamendes@gmail.com (S.M.); fonseca.ja@gmail.com (J.A.F.); 2Department of Community Medicine, Information and Health Decision Sciences (MEDCIDS), Faculty of Medicine, University of Porto, 4200-319 Porto, Portugal; ferreirademagalhaes@gmail.com; 3Department of Cardiovascular and Respiratory Sciences, Porto Health School, Polytechnic Institute of Porto, 4200-072 Porto, Portugal; 4Department of Women’s and Children’s Health, Paediatric Research, Uppsala University, SE-751 05 Uppsala, Sweden; 5Allergy Unit, CUF Porto Hospital and Institute, 4100-180 Porto, Portugal; magnacorreia1@gmail.com (M.A.-C.); luisaraujo78@gmail.com (L.A.); 6Serviço de Pediatria, Hospital Pedro Hispano, Unidade Local de Saúde de Matosinhos, 4464-513 Matosinhos, Portugal; jccidraisrodrigues@hotmail.com (J.C.C.R.); joana.teixeiracarvalho@gmail.com (J.C.); cristina.lopes.abreu@gmail.com (C.L.); 7Serviço de Pediatria, Hospital Senhora da Oliveira, 4835-044 Guimarães, Portugal; agcosta40@gmail.com (A.C.); armandinapf@gmail.com (A.S.); 8Serviço de Pediatria, Centro Materno Infantil do Norte, Centro Hospitalar Universitário do Porto, 4099-001 Porto, Portugal; mfernandateixeira59@gmail.com; 9Serviço de Imunoalergologia, Hospital do Divino Espírito Santo, 9500-370 Ponta Delgada, Portugal; rodrigosralves@gmail.com; 10Unidade de Imunoalergologia, Hospital do Divino Espírito Santo, 9500-370 Ponta Delgada, Portugal; aninhasnet@gmail.com; 11Departamento de Pediatria, Hospital de Santa Maria, Centro Hospitalar de Lisboa Norte, 1649-035 Lisboa, Portugal; rmfernandes@campus.ul.pt (R.M.F.); rosariotferreira@sapo.pt (R.F.); amargaretemmendes@gmail.com (A.M.); 12Serviço de Imunoalergologia, Hospital de Dona Estefânia, Centro Hospitalar Universitário de Lisboa Central, 1150-199 Lisboa, Portugal; pleiriapinto@gmail.com (P.L.P.); nuno.neuparth@nms.unl.pt (N.N.); 13Pathophysiology, CHRC/CEDOC, High Burden and High Mortality Diseases Thematic Line Coordinator, Nova Medical School, 1150-190 Lisboa, Portugal; 14Serviço de Pediatria, Unidade Hospitalar de Famalicão, Centro Hospitalar do Médio Ave, 4780-371 Vila Nova de Famalicão, Portugal; diana.bordalo@gmail.com; 15Serviço de Imunoalergologia, Centro Hospitalar e Universitário de Coimbra, 3000-075 Coimbra, Portugal; flcosta@netcabo.pt; 16Serviço de Pediatria, Hospital de São Teotónio, Centro Hospitalar Tondela–Viseu, 3504-509 Viseu, Portugal; mariajosecalix@gmail.com; 17Unidade de Saúde Familiar Progresso e Saúde, ACeS Baixo Mondego, 3060-716 Tocha, Portugal; tani_ferreira@hotmail.com; 18Serviço de Imunoalergologia, Unidade I, Centro Hospitalar Vila Nova de Gaia/Espinho, 4434-502 Vila Nova de Gaia, Portugal; joanarqueirosg@gmail.com; 19Servicio de Alergia, Complejo Hospitalario Universitario de Santiago, 15706 Santiago De Compostela, Spain; carmen.vidal.pan@sergas.es; 20Serviço de Imunoalergologia, Centro Hospitalar Universitário de São João, 4200–319 Porto, Portugal; mariajoaosvasconcelos@gmail.com; 21Imunoalergologia, Grupo HPA Saúde, 8500-322 Portimão, Portugal; pedrotiagosilva@gmail.com; 22Imunoalergologia, Hospital Privado de Alfena, Trofa Saúde, 4445-243 Alfena, Portugal; zeferrazdeo@gmail.com; 23Serviço de Imunoalergologia, Hospital Infante D. Pedro, Centro Hospitalar Baixo Vouga, 3814-501 Aveiro, Portugal; anamorete@gmail.com; 24Serviço de Pneumologia, Hospital São Pedro de Vila Real, Centro Hospitalar De Trás-Os-Montes E Alto Douro, 5000-508 Vila Real, Portugal; csmnspinto@gmail.com; 25Serviço de Imunoalergologia, Centro Hospitalar Universitário do Algarve, 8000-386 Portimão, Portugal; nsantos.alergia@gmail.com; 26Serviço de Pneumologia, Hospitais da Universidade de Coimbra, 3000-076 Coimbra, Portugal; cl_loureiro@hotmail.com; 27Serviço de Pneumologia, Centro Hospitalar e Universitário de Coimbra, 3000-075 Coimbra, Portugal; ana.arrobas@gmail.com; 28Serviço de Imunoalergologia, Centro Hospitalar Universitário do Porto, 4099-001 Porto, Portugal; maluis234@gmail.com; 29Serviço de Imunoalergologia, Hospital Amato Lusitano, Unidade Local de Saúde de Castelo Branco, 6000-085 Castelo Branco, Portugal; clozoya@fcsaude.ubi.pt; 30Imunologia Básica e Clínica, Faculdade de Medicina, Universidade do Porto, 4200-319 Porto, Portugal; 31Unidade de Saúde Familiar Terras de Azurara, ACES Dão Lafões, 3530-113 Mangualde, Portugal; francisca.cardia@gmail.com; 32Departamento de Pediatria, Serviço de Pediatria Ambulatória, Centro Hospitalar e Universitário de Coimbra, 3000-075 Coimbra, Portugal; carlachavesloureiro@gmail.com; 33Serviço de Pneumologia, Hospital Nossa Senhora do Rosário, Centro Hospitalar Barreiro Montijo, 2834-003 Barreiro, Portugal; raquelpaulinetticamara@gmail.com; 34UCSP Dr. Arnaldo Sampaio, ACES Pinhal Litoral, 2419-014 Leiria, Portugal; ines_b_vieira@hotmail.com; 35USF Cuidarte, Unidade Local de Saúde do Alto Minho, 4925-083 Portuzelo, Portugal; sofia.carla.silva@gmail.com; 36Unidade de Saúde Familiar João Semana, ACeS Baixo Vouga, 3880-225 Ovar, Portugal; euriko7@gmail.com; 37Unidade de Saúde Familiar Mondego, ACES Baixo Mondego, 3045-059 Coimbra, Portugal; natalinarodrigues89@gmail.com

**Keywords:** asthma, adolescents, phenotypes, clustering, longitudinal studies, latent class analysis

## Abstract

We aimed to identify persistent asthma phenotypes among adolescents and to evaluate longitudinally asthma-related outcomes across phenotypes. Adolescents (13–17 years) from the prospective, observational, and multicenter INSPIRERS studies, conducted in Portugal and Spain, were included (*n* = 162). Latent class analysis was applied to demographic, environmental, and clinical variables, collected at a baseline medical visit. Longitudinal differences in clinical variables were assessed at a 4-month follow-up telephone contact (*n* = 128). Three classes/phenotypes of persistent asthma were identified. Adolescents in class 1 (*n* = 87) were highly symptomatic at baseline and presented the highest number of unscheduled healthcare visits per month and exacerbations per month, both at baseline and follow-up. Class 2 (*n* = 32) was characterized by female predominance, more frequent obesity, and uncontrolled upper/lower airways symptoms at baseline. At follow-up, there was a significant increase in the proportion of controlled lower airway symptoms (*p* < 0.001). Class 3 (*n* = 43) included mostly males with controlled lower airways symptoms; at follow-up, while keeping symptom control, there was a significant increase in exacerbations/month (*p* = 0.015). We have identified distinct phenotypes of persistent asthma in adolescents with different patterns in longitudinal asthma-related outcomes, supporting the importance of profiling asthma phenotypes in predicting disease outcomes that might inform targeted interventions and reduce future risk.

## 1. Introduction

Asthma is one of the most common chronic diseases in children worldwide [1]. Despite advances and changes in guidelines, there is no known treatment for asthma and the main goal, which is to achieve disease control, remains challenging [2].

Children with uncontrolled asthma need to use asthma medication more frequently and are more likely to use healthcare services due to their asthma, with an increase in unscheduled medical visits, and hospital admissions [3,4,5]. Likewise, adolescents have poorer outcomes and worse adherence due to a lack of self-management skills and insufficient health literacy knowledge [6].

Classification and understanding of heterogeneous asthma phenotypes are the starting point to establish individualized management plans [7] and might lead to improvements in asthma control. Recent studies using data-driven methods provided novel insights into meaningful and accurate asthma phenotypes based on real-life data [8,9,10].

However, phenotypic characterizations of adolescents with asthma are very limited, and longitudinal studies that potentially predict long-term outcomes and personalizing treatments are scarce. Moreover, variables collected longitudinally are of extreme importance to evaluate the stability of any derived phenotypes and to further validate them, particularly in unselected subjects with asthma from the general population. The INSPIRERS studies assessed adherence to inhaled medication among adolescents with persistent asthma, collecting real-life data over time, with the potential to support and enable patient-centered care and research [11,12].

Therefore, this study aims to identify persistent asthma phenotypes among adolescents and to evaluate longitudinally asthma-related outcomes across phenotypes.

## 2. Materials and Methods

### 2.1. Study Population

We performed a secondary analysis of data from adolescents (13–17 years) enrolled in the prospective, observational, and multicenter INSPIRERS studies, which have been described previously [11,13]. Briefly, the INSPIRERS studies assessed adherence to asthma inhalers among adolescents and adults with persistent asthma. During a face-to-face baseline medical visit, patients were invited by their physicians to participate in the study, and a convenience sample was obtained, between March 2018 and January 2020, at 30 primary and secondary care centers from Portugal and Spain.

The study protocol was approved by the Ethics Committee of all participating centers. Before enrolment in the study, adolescents signed an assent form and a written consent form from the parent(s) or legal guardian(s) was also obtained. In this study, we followed the STROBE statement for reporting of observational studies [14].

### 2.2. Study Design and Procedures

Participants were eligible for this secondary analysis if they: (1) had a previous medical diagnosis of persistent asthma, (2) were between 13 and 17 years old, and (3) had an active prescription for a daily inhaled controller medication for asthma. Exclusion criteria included a diagnosis of a chronic lung disease other than asthma or other chronic condition with possible interference with the study aims.

Participants completed a face-to-face baseline visit (T0), where physicians reported patients’ asthma treatment; an assessment of asthma control according to the Global Initiative for Asthma (GINA) [2]; last reported value of percent predicted Forced Expiratory Volume in the first second (FEV_1_); number of exacerbations in the past year (defined as episodes of progressive increase in shortness of breath, cough, wheezing, and/or chest tightness, requiring a change in maintenance therapy [15]) and of unscheduled medical visits in the past year (consultations at primary care, specialist’s office, hospitalizations, or emergency department).

A sociodemographic and clinical questionnaire was administered to the participants, including height and weight, an assessment of asthma control during the previous 4 weeks by Control of Allergic Rhinitis and Asthma Test (CARAT) questionnaire [16]. CARAT total score (CARAT-T) is calculated by summing up the scores of all 10 questions, resulting in a range of 0–30 points. CARAT has two domains: upper airways (CARAT-UA, range: 0–12) and lower airways (CARAT-LA, range: 0–18). A score > 24 on CARAT-T, >8 on CARAT-UA, and ≥16 on CARAT-LA were used to define control regarding total, upper, and lower airways symptoms, respectively [17].

Participants were interviewed by phone at 1 week (T1), 1 month (T2), and 4 months (T3) after the face-to-face baseline visit, and their asthma control was again assessed using CARAT. Also, exacerbations and unscheduled medical visits in the past 4 months were recorded at T3.

### 2.3. Statistical Analysis

Latent class analysis (LCA) was applied to 10 variables easily collected in medical visits both primary and secondary care, collected at T0: sex (Male/Female), asthma symptom’s onset before the age of 6 years (Yes/No), presence of comorbidities (Yes/No, yes if at least one of the following physician-reported: atopic dermatitis, rhinitis, and rhinosinusitis), body mass index (BMI) ≥ 85th percentile [18] (Yes/No), exposure to environmental tobacco smoke (ETS) (Yes/No), Pre-BD FEV_1_ < 80% (Yes/No), CARAT-UA and CARAT-LA (controlled/uncontrolled), ≥1 exacerbation (Yes/No), and ≥1 unscheduled healthcare visits (Yes/No). The optimal number of classes resulting from the variables was determined by evaluating k classes versus k-1 classes sequentially, until adding a class no longer significantly improved the model, measured by Lo-Mendell-Rubin-adjusted likelihood ratio test. The best model was determined by the largest entropy and the lowest Bayesian information criteria (BIC) values [19]. Longitudinal differences in clinical variables (CARAT, exacerbations, and unscheduled medical visits) were assessed by comparison between baseline, T1, T2, and T3 data. The ratio of the number of exacerbations/unscheduled healthcare visits per month was also calculated.

Categorical variables were presented as absolute frequencies and proportions. Continuous variables were presented according to their distributions: mean and standard deviation (sd) or median (percentile 25–percentile 75: P25–P75). Group comparisons were performed by independent *t*-test, Mann–Whitney, and Kruskal–Wallis tests for continuous variables or Chi-square test for categorical variables. Multiple testing was conducted using the Bonferroni correction when needed. Longitudinal changes in each variable were assessed using a generalized linear model with pairwise comparisons of means, to analyze the differences between the classes over time.

Statistical analyses were performed using IBM SPSS Statistics V.26.0 (IBM Corporation, Armonk, NY, USA), and MPlus 6.12 (Muthén & Muthén, Los Angeles, CA, USA) was used to conduct LCA analysis. Plots were created using GraphPad Prism V.6.0 (GraphPad Software, La Jolla, CA, USA). The level of significance was set at 0.05.

## 3. Results

### 3.1. Characteristics of the Study Population

The sample consisted of 162 adolescents with a median (P25–P75) of 15 (14–16) years, 128 of which (78%) completed the three follow-up interviews. Baseline characteristics are shown in Table 1.

There were no significant differences between sex and baseline characteristics, except in: GINA asthma control, with females having significantly more proportion of uncontrolled asthma than males (*p* = 0.033); and CARAT-T and CARAT-LA, with females presenting lower scores, indicating more symptoms and poorer control (Table 1). Moreover, most of the adolescents used a single inhaler (64%) and ICS/LABA was commonly used as a controller asthma medication (Table 1).

### 3.2. Classes of Persistent Asthma

A three-class model was selected as the best solution for these data (Table 2), with a significantly better fitting than a two-class model (*p* = 0.001), and a non-significantly different fit from a four-class model (*p* = 0.195). Furthermore, the entropy of the three-class model was 0.851, a very good overall certainty in classification, and this was the best fit for phenotype identification as it had the lowest BIC value with minimal loss of entropy.

Adolescents in class 1 (*n* = 87) were equally distributed regarding sex, with most of them having at least one comorbidity (72%). Although only 17% had FEV_1_ < 80%, they were highly symptomatic at baseline, with a high proportion having unscheduled healthcare visits and exacerbations in the past year (57% and 97%, respectively).

Class 2 (*n* = 32) was characterized by a predominance of females (80%), a higher proportion of overweight/obese adolescents, and all had FEV_1_ above 80%. However, this class had the highest proportion of adolescents with uncontrolled upper/lower airways symptoms, with a low proportion of exacerbations (11%) and without unscheduled healthcare visits.

Class 3 (*n* = 43) included mostly males, half of whom had uncontrolled upper airway symptoms and the majority having controlled lower airways symptoms. Similar to class 2, in class 3 we observed a very low proportion of exacerbations and unscheduled healthcare visits.

### 3.3. Longitudinal Assessment of Latent Classes

Of the 162 adolescents included in the baseline visit (T0), 139, 136, and 128 adolescents respectively completed the T1, T2, and T3 follow-up phone interview. Figure 1 and Figure 2 show the longitudinal changes in CARAT scores (across T0, T1, T2, and T3) and in the number of exacerbations/healthcare unscheduled per month (between T0 and T3).

Adolescents in class 1 presented an increase in CARAT scores across all time points (Figure 1) and had the highest mean number of exacerbations and unscheduled healthcare visits/month, both at baseline (0.25 and 0.10, respectively) and follow-up (0.44 and 0.14, respectively) (Figure 2).

Class 2 at T3 follow-up presented a significant increase in the proportion of adolescents with controlled lower airways symptoms (*p* < 0.001), but also a significant increase in the number of exacerbations/month (mean difference = 0.22, *p* = 0.009). The number of unscheduled healthcare visits also increased, although non-significantly (mean difference = 0.14, *p* = 0.10).

In class 3, although most participants kept a high proportion of lower airway symptom control at follow-up, there was a significant increase in the mean number of exacerbations/month (mean difference = 0.27, *p* = 0.015).

## 4. Discussion

In our study, we identified three distinct phenotypes (classes) of persistent asthma in adolescents that presented different patterns in longitudinal asthma-related outcomes. These classes differed significantly concerning airways symptoms, exacerbations, and the need for unscheduled healthcare visits, particularly in their longitudinal changes at follow-up.

To identify the phenotypic heterogeneity over time in children and adolescents with asthma, clustering techniques have been applied to broad cohorts, aiming to describe and monitor asthma phenotypes [8,9,10,20]. However, the present study focused on a specific population, adolescents with persistent asthma, and used the obtained phenotypes as a starting point for the follow-up of these asthma phenotypes, enabling the assessment of the trajectories of asthma control.

In our study, both at cross-sectional and longitudinal levels, the obtained classes of adolescents with persistent asthma are clinically reasonable. Class 1 was a troublesome exacerbation-prone asthma phenotype with a high proportion of uncontrolled disease and more unscheduled healthcare visits, while class 3 was the mildest phenotype of adolescents with persistent asthma, being similar to other phenotypes found in the literature among this population [8,21,22,23]. Class 2, compared to the other classes, included predominantly female adolescents, who were more frequently obese and had later-onset asthma symptoms (>6 years-old), with a high proportion of uncontrolled disease, but with few self-reported exacerbations/unscheduled healthcare visits. Class 2 had the lowest CARAT scores (T, UA, and LA) at baseline, and significantly improved after four months, even with higher CARAT LA scores than class 1. Moreover, among class 2, there was a meaningful change in CARAT T score (improvement of 5 points), above the minimal clinically important difference reported for CARAT questionnaire (3.5 points) [24], although it was not further assessed in confirmatory studies, namely in adolescents. To our knowledge, this is the first time that this phenotype is described in adolescents with persistent asthma; however, additional studies that include more participants and more comprehensive variables are needed to validate it.

All classes reported an increase in the number of exacerbations per month in the follow-up, and this was significant in classes 2 and 3. A possible effect of the seasonality of the exacerbations might be a cause of this increase; however, when interpreting these results, one should bear in mind that a possible memory bias could explain this increase, as participants were likely better able to remember exacerbations during the past 4 months, compared to the first assessment at baseline (focused on the past 12 months). Also, exacerbations at T0 were physician-reported and at T3 were self-reported, without clinical validation. Specifically, class 3 was composed mainly of male adolescents that, although keeping symptom control, reported a significant increase in exacerbations. This suggests an underestimation of their airway symptoms that, together with the fact that adolescents often wish to take charge of their health and/or stop the medication due to stigma [6,25], might lead to poorer outcomes. Moreover, these findings support not only the importance of continuously monitoring childhood asthma to reduce the impact of this disease [1] but also that, in older children, asthma monitoring should be based both on symptomatic patterns and in objective variable expiratory airflow limitation.

Indeed, evaluation of symptoms plays a key role in asthma diagnosis and management [2]; however, diagnostic tests, such as lung function tests, are also important for the diagnosis and assessment of disease. In our study, FEV_1_ was predominantly normal (>80%) in classes 1 and 2, which was in line with the findings of Lee et al. [8] that described that in some clusters/phenotypes of children and adolescents with persistent asthma, lung function was normal, particularly in those with a lower proportion of atopy. However, because we did not assess atopy/sensitization, a direct comparison is limited, and this should be further studied.

Similar to our findings, it has already been described that children with uncontrolled asthma need to use more asthma medication and are more likely to use healthcare services for asthma, especially unscheduled medical visits and hospital admissions [4,26]. This points to the importance of improving knowledge on asthma control in the long-term, particularly among adolescents. Innovative mobile health (mHealth) applications could be a means to approach this issue, as smartphones are now widespread [27] and apps are very appealing to inspire behavior changes, through gamification and social support, particularly in this age range [11,28].

There are many opportunities regarding longitudinal real-life data combined with mHealth applications, such as InspirerMundi [29] and MASK [30], which are becoming increasingly popular among physicians, patients, and the general public. However, future studies that combine hypothesis-independent clustering and real-life data extracted from mHealth applications applied to asthma diagnosis and management are needed.

There are limitations in this study that should be acknowledged. First, the INSPIRERS studies cohort is not a random sample of individuals with persistent asthma, and inclusion was based on the presence of physician-diagnosed asthma without the need for objective measurements, such as lung function tests, to support the diagnosis. However, our findings must be further validated to be generalized to the greater population of adolescents with persistent asthma. Second, the lack of data regarding atopy and viral infections should be considered in the interpretation of our findings, as they play a major role in asthma control and management in later childhood and adulthood [31]. Third, as with any data-driven clustering, there are limitations in the interpretation of derived classes as being a true set of clinically meaningful subgroups [32]; however, no clustering/group effect on the asthma-related outcomes was observed (data not shown). Also, the choice of the variables included in the LCA model was based on parameters being easily reported at a medical consultation, both in primary and secondary care, and that could potentially be useful in disease management; however, the inclusion/replacement of other variables representing different disease domains, such as adherence to treatment, medication and inflammatory biomarkers, should be explored. Finally, the cluster stability could not be assessed because of the limited number of adolescents in T3; however, we aimed to monitor the baseline latent classes concerning longitudinal asthma-related outcomes.

## 5. Conclusions

We have identified three distinct phenotypes of persistent asthma in adolescents that presented different patterns in longitudinal asthma-related outcomes, supporting the importance of profiling asthma phenotypes in predicting disease outcomes, which might inform targeted interventions and reduce future risk.

## Figures and Tables

**Figure 1 ijerph-18-01015-f001:**
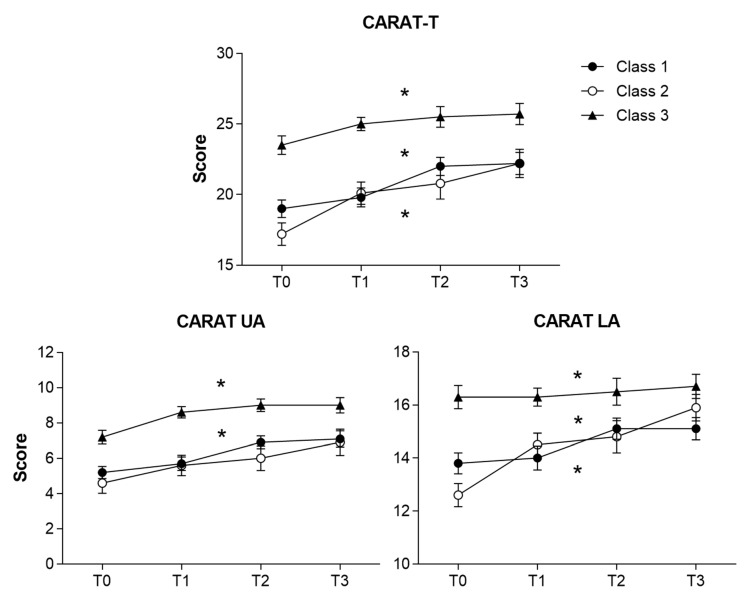
Longitudinal changes in the three CARAT scores (T, UA, and LA), in classes 1 (*n* = 89), class 2 (*n* = 31) and class 3 (*n* = 43). Error bars indicating standard error of the mean. CARAT: Control of Allergic Rhinitis and Asthma Test; T0: baseline assessment; T: total; UA: Upper airways; LA: Lower airways. * significant overall longitudinal changes (*p* < 0.05). T0: baseline assessment; T1: 1-week; T2: 1-month; T3: 4-month follow-up.

**Figure 2 ijerph-18-01015-f002:**
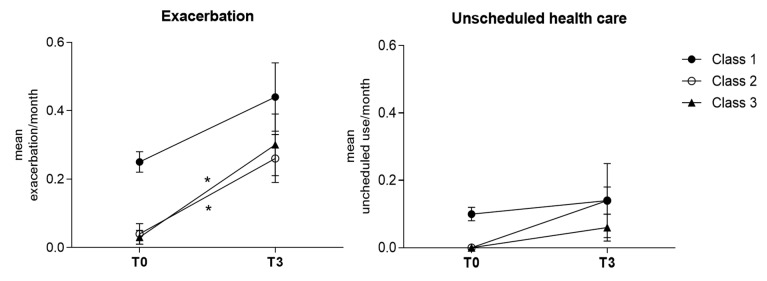
Longitudinal changes in the rate of exacerbations and unscheduled healthcare visits per month, among the three classes. Error bars indicating standard error of the mean. Number of subjects with exacerbations: class 1 (T0: *n* = 81; T3: *n* = 33), class 2 (T0: *n* = 3; T3: n = 10) and class 3 (T0: *n* = 3; T3: *n* = 16). Number of subjects with unscheduled healthcare visits: class 1 (T0: *n* = 49; T3: *n* = 12), class 2 (T0: *n* = 0; T3: *n* = 2) and class 3 (T0: *n* = 0; T3: *n* = 3). * significant for *p* < 0.01. T0: baseline assessment; T3: 4-month follow-up.

**Table 1 ijerph-18-01015-t001:** Characteristics of adolescents at the time of enrolment (T0), according to sex (*n* = 162).

Characteristics	Total *n* = 162	Male *n* = 83	Female *n* = 79	*p* Value ^2^
Age, median (P25–P75)	15 (14–16)	14 (13–16)	15 (14–16)	0.061
Age of Symptom’s onset, median (P25–P75)	6 (3–10)	5.5 (3–10)	7.0 (3–10)	0.538
BMI classification, *n* (%):				
Underweight (<5th percentile)	3 (2)	1 (1)	2 (3)	0.537
Healthy weight (5th–85th percentile)	117 (76)	62 (80)	55 (73)	0.370
Overweight (85th–95th percentile	21 (14)	9 (11)	12 (16)	0.443
Obesity (≥95th percentile)	12 (8)	6 (8)	6 (8)	0.944
Pre-BD FEV_1_% predicted, mean (sd)	95.6 (16.2)	95.0 (16.8)	95.9 (15.7)	0.735
Exposure to ETS, *n* (%)	71 (47)	36 (43)	35 (47)	0.888
Comorbidities ^1^, *n* (%)	111 (68)	59 (64)	58 (73)	0.190
Single inhaler, *n* (%)	104 (64)	57 (69)	47 (59)	0.223
Inhaled medication, *n* (%):				
ICS/LABA	114 (72)	57 (70)	57 (74)	0.608
SABA	52 (33)	22 (27)	30 (39)	0.115
ICS	45 (28)	24 (30)	21 (27)	0.743
LAMA	4 (2)	2 (2)	2 (3)	0.959
Asthma control (GINA), *n* (%)				
Controlled	74 (46)	42 (51)	32 (41)	0.223
Partly controlled	62 (38)	33 (40)	29 (37)	0.737
Uncontrolled	25 (16)	8 (10)	17 (22)	**0.033**
CARAT-T, median (P25–P75)	21 (16–24)	22 (17–25)	18 (14–23)	**0.010**
CARAT-UA, median (P25–P75)	6 (3–8)	6 (4–8)	5 (2–7)	0.101
CARAT-LA, median (P25–P75)	15 (12–17)	16 (13–18)	14 (11–16)	**0.003**
Exacerbations, *n* (%)	91 (58)	44 (55)	47 (61)	0.443
Unscheduled healthcare visits, *n* (%)	39 (31)	24 (34)	15 (27)	0.395

^1^ Comorbidities included rhinitis, rhinosinusitis, or atopic dermatitis. ^2^ Independent *t*-test, Mann–Whitney, or Chi-square tests. In bold are statistically significant *p* value at <0.05. BMI: body mass index; CARAT: Control of Allergic Rhinitis and Asthma Test; T: Total; UA: Upper airways; LA: Lower airways; GINA: Global Initiative for Asthma, Pre-BD: Pre-bronchodilator; FEV_1_: Forced expiratory volume in one second; ICS: inhaled corticosteroid; LABA: Long-acting beta-agonists; SABA: Short-acting beta agonists; LAMA: Long-acting muscarinic antagonists.

**Table 2 ijerph-18-01015-t002:** Results of latent class analysis at the time of enrollment (T0).

Variables	Class 1 *n* = 87	Class 2 *n* = 32	Class 3 *n* = 43	1 vs. 2 *p* Value ^1^	1 vs. 3 *p* Value ^1^	2 vs. 3 *p* Value ^1^
Male	44 (50)	6 (20)	33 (76)	**0.004**	**0.006**	**<0.001**
BMI ≥ 85th percentile	16 (19)	8 (25)	8 (19)	0.637	0.925	0.625
Exposure to ETS	41 (48)	11 (39)	18 (47)	0.413	0.933	0.513
Comorbidities	63 (72)	17 (57)	32 (76)	0.797	0.379	0.647
Symptom’s onset, before age of 6 years	45 (52)	10 (33)	20 (50)	0.073	0.808	0.163
Pre-BD FEV_1_ < 80%	11 (17)	0 (0)	8 (23)	**0.041**	0.517	**0.022**
CARAT-T, uncontrolled	70 (81)	30 (97)	23 (56)	**0.035**	**0.003**	**<0.001**
CARAT-UA, uncontrolled	71 (82)	28 (90)	26 (63)	0.292	**0.023**	**0.011**
CARAT-LA, uncontrolled	52 (60)	31 (100)	6 (15)	**<0.001**	**<0.001**	**<0.001**
Exacerbations, last year	81 (97)	3 (11)	3 (8)	**<0.001**	**<0.001**	0.669
Unscheduled healthcare visits, last year	49 (57)	0 (0)	0 (0)	**<0.001**	**<0.001**	0.999

Numbers are presented as *n* (%). Three participants were excluded due to missing data on the main variables in the model. ^1^ Chi-square test. In bold are statistically significant *p*-value at <0.05. BMI: body mass index; ETS: environmental tobacco smoke; Pre-BD: pre-bronchodilator; FEV_1_: Forced expiratory volume in one second; CARAT: Control of Allergic Rhinitis and Asthma Test; UA: upper airways; LA: Lower airways.

## Data Availability

The data presented in this study are available on request from the corresponding author. The data are not publicly available due to ethical restrictions.

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
