# Peer review of "Profiling Persistent Asthma Phenotypes in Adolescents: A Longitudinal Diagnostic Evaluation from the INSPIRERS Studies"

_ijerph, 2021, doi:10.3390/ijerph18031015_

Round 1

Reviewer 1 Report

GENERAL

The authors identified 3 latent classes of asthma phenotypes in a sample of 162 adolescents and assessed asthma and rhinitis control, exacerbations, and unscheduled healthcare visits over 4-months follow-up in each class. The study is interesting and has elements of originality since the literature is scarce in adolescents. The text reads well and is sufficiently concise. There are some issues that should be considered.

SPECIFIC

1.       There is no justification for the choice of the 10 variables in the latent class analysis. Apparently, other variables, possibly more relevant, were available (type of treatment, dose of inhaled corticosteroids, adherence to treatment, smoking habit). Moreover, FEV1/FVC ratio would be a more appropriate index of airflow limitation, than FEV1 alone which is a more global index of functional impairment, either obstructive and restrictive, and is influenced by the growth of lungs in the age range of the sample.

2.       Table 2. It would be more informative to provide statistics on which variables differed significantly between classes.

3.       The interpretation of longitudinal changes needs to consider drug treatment at baseline (T0) and at follow-up (T3). It is peculiar that there is an overall trend of less controlled disease and more exacerbation at the 4-months assessment, since patients involved in a study generally tend to improve. A possible effect of the seasonality (Discussion, pag 8) is unlikely, since the subjects were not recruited at the same time.

4.       The conclusion that the phenotypes ‘presented different trajectories in asthma-related outcomes’ does not seem clearly supported by the longitudinal data shown in Figures 1 and 2, where all classes exhibit increase of CARAT scores and rate of exacerbation.

Reviewer 2 Report

  1. The clusters were based solely on clinical variables; how about other key variables such as peripheral blood eosinophil level, exhaled nitric oxide, allergic sensitizations results? Do you think  cluster analysis based solely on clinical variables would be clinically meaningful? Please provide support to the current study design and/or explain in the study limitations.
  2. Patients included in the analysis had a physician diagnosis of "persistent" asthma; however, there is no objective confirmation of their asthma diagnosis? How would you explain the high proportion of normal FEV1 in the majority of your study participants? Please consider adding this to the study limitations as they may not be very suitable to answer the key study question.
  3. Please provide rational for combining rhinitis, sinusitis and atopic dermatitis under one category? The contribution of these variables to asthma phenotyping and control may not equal and therefore, I would suggest re-visting the cluster analysis separating these variables.
  4. Can you please specify the clinical meaningful change in CART score, a change of 1 or 3 may not be clinically significant and readers would appreciate the results further with such information. 
  5. Can you please report effect size (mean difference or the mean for each group) to add better context to your findings and not simply the P values?
  6. In the figures 1 and 2 (These are key to your study question)
    1. can you also please plot the Standard Error of the Mean (SEM)
    2. You adopted a linear model (explain rational for selecting this method, is there missing data point?)
    3. In the model; there are 2 factors (time and cluster and there is interaction between these 2 factors). You have examined the change over time. Can you also report the difference between the clusters (i.e., does clustering has a significant effect over the change in CART score/exacerbation rate/healthcare utilization over time?. At this point, you only answered whether time affected the scores in these clusters, but as you indicated in the discussion, this may be related to seasonality or other unidentified confounders.

Round 2

Reviewer 1 Report

The authors adequately considered the comments of the reviewer and revised the manuscript accordingly. There are no further comments.